# Human Lung Cancer (A549) Cell Line Cytotoxicity and Anti-*Leishmania major* Activity of *Carissa macrocarpa* Leaves: A Study Supported by UPLC-ESI-MS/MS Metabolites Profiling and Molecular Docking

**DOI:** 10.3390/ph15121561

**Published:** 2022-12-14

**Authors:** Mohamed A. A. Orabi, Omaish Salman Alqahtani, Bandar A. Alyami, Ahmed Abdullah Al Awadh, El-Shaymaa Abdel-Sattar, Katsuyoshi Matsunami, Dalia I. Hamdan, Mohamed E. Abouelela

**Affiliations:** 1Department of Pharmacognosy, College of Pharmacy, Najran University 1988, Najran 66454, Saudi Arabia; 2Department of Pharmaceutical Chemistry, College of Pharmacy, Najran University 1988, Najran 66454, Saudi Arabia; 3Department of Clinical Laboratory Sciences, Faculty of Applied Medical Sciences, Najran University 1988, Najran 66454, Saudi Arabia; 4Department of Microbiology and Immunology, Faculty of Pharmacy, South Valley University, Qena 83523, Egypt; 5Department of Pharmacognosy, Graduate School of Biomedical and Health Sciences, Hiroshima University, 1-2-3 Kasumi, Minami-Ku, Hiroshima 734-8553, Japan; 6Department of Pharmacognosy and Natural Products, Faculty of Pharmacy, Menoufia University, Shibin Elkom 32511, Egypt; 7Department of Pharmacognosy, Faculty of Pharmacy, Al-Azhar University, Assiut-Branch, Assiut 71524, Egypt

**Keywords:** *Carissa macrocarpa*, cytotoxicity, A549, *Leishmania major*, UPLC-ESI-MS/MS, HDAC6, PDK3, molecular docking

## Abstract

Lung cancer and cutaneous leishmaniasis are critical diseases with a relatively higher incidence in developing countries. In this research, the activity of *Carissa macrocarpa* leaf hydromethanolic extract and its solvent-fractions (*n*-hexane, EtOAc, *n*-butanol, and MeOH) against the lung adenocarcinoma cell line (A549) and *Leishmania major* was investigated. The MeOH fraction exhibited higher cytotoxic activity (IC_50_ 1.57 ± 0.04 μg/mL) than the standard drug, etoposide (IC_50_ 50.8 ± 3.16 μg/mL). The anti-*L. major* results revealed strong growth inhibitory effects of the EtOAc fraction against *L. major* promastigotes (IC_50_ 27.52 ± 0.7 μg/mL) and axenic amastigotes (29.33 ± 4.86% growth inhibition at 100 μg/mL), while the butanol fraction exerted moderate activity against promastigotes (IC_50_ 73.17 ± 1.62), as compared with miltefosine against promastigotes (IC_50_ 6.39 ± 0.29 μg/mL) and sodium stibogluconate against axenic amastigotes (IC_50_ 22.45 ± 2.22 μg/mL). A total of 102 compounds were tentatively identified using UPLC-ESI-MS/MS analysis of the total extract and its fractions. The MeOH fraction was found to contain several flavonoids and flavan-3-ol derivatives with known cytotoxic properties, whereas the EtOAc fractions contained triterpene, hydroxycinnamoyl, sterol, and flavanol derivatives with known antileishmanial activity. Molecular docking of various polyphenolics of the MeOH fraction with HDAC6 and PDK3 enzymes demonstrates high binding affinity of the epicatechin 3-*O*-β-D-glucopyranoside and catechin-7-*O*-β-D-glucopyranoside toward HDAC6, and procyanidin C2, procyanidin B5 toward PDK3. These results are promising and encourage the pursuit of preclinical research using *C. macrocarpa*’s MeOH fraction as anti-lung cancer and the EtOAc fraction as an anti-*L. major* drug candidates.

## 1. Introduction

Cancer is a leading cause of death worldwide, accounting for nearly 10 million deaths in 2020 [1]. Lung cancer is the second most frequent cancer and the major cause of cancer-related death [1]. The Middle East and North Africa have a higher incidence rate of lung cancer, exacerbated by the region’s heavy tobacco use, which accounts for 85% of all instances. This condition warrants increased efforts to discover an urgent treatment [2,3]. Despite chemotherapeutic drugs’ high efficacy, many patients experience serious chemotherapy-induced side effects [4]. Several attempts have been made to investigate new agents that can work synergistically with chemotherapy or as an adjuvant to reduce their side effects.

Natural remedies have drawn a lot of interest in the fight against cancer because they are thought to be more cellular-friendly, more focused on their targets, and less harmful to healthy cells [5]. There is evidence that natural product-derived anticancer medications utilize alternative strategies for causing cell death [6].

Cutaneous leishmaniasis (CL), a parasitic disease caused by *Leishmania major*, is spread by the bites of specific species of sandflies and affects up to 1.5 million individuals across 89 nations [7]. The disease is endemic in the Kingdom of Saudi Arabia (KSA); the majority of incidents occur in Al-Qaseem, Riyadh, Al-Hassa, Aseer, Ha’il, and Al-Madinah, which, despite efforts by health officials, remains a significant public health problem [8].

The awful side effects and high expense of therapeutic medications make treating CL difficult. Therefore, as a part of our continuous efforts [9,10] to exploit local plants in the KSA to overcome lung cancer and CL, we phytochemically and pharmacologically investigated *Carissa macrocarpa* (Eckl.) DC (Apocynaceae) leaves extract. *C. macrocarpa*, the Natal plum, is a widespread ornamental shrub or small tree characterized by its enormous lush, green, and persistent leaves, white star-shaped blooms, and edible oval fruits [11,12]. Phytochemically, it is characterized by the production of flavonoids, saponins, sterols, terpenes, anthraquinone, tannins, lignans, and fatty acids [11,12]. Pharmacological testing on the hydroethanolic extract from the fruits and leaves of *C. macrocarpa* revealed growth inhibition of the cervical (HeLa), non-small cell lung carcinoma (NCI-H460), and breast (MCF-7) cancer cell lines [11,12,13]. Furthermore, various *Carissa* species have demonstrated potent anti-infectious disease activities, including anthelmintic, antiplasmodial, antibacterial, and antiviral effects [14,15].

This study aims to evaluate the in vitro activity of the *C. macrocarpa* leaf extract and fractions against lung cancer cell lines (A549) and the *L. major* promastigote and axenic amastigote. In addition, ultra-performance liquid chromatography-electrospray ionization-mass spectrometry (UPLC-ESI-MS/MS) analysis was used to qualitatively examine the metabolites of the *C. macrocarpa* leaves’ total extract and its polar solvent fractions (EtOAc fr., n-butanol fr., and MeOH fr.). It is worth noting that it is now widely accepted that pyruvate dehydrogenase kinase 3 (PDK3) and histone deacetylase 6 (HDAC6) inhibitors can be used to treat lung cancer [16,17]. Several flavanols and flavonoids derivatives identified using UPLC-ESI-MS/MS analysis of the potently cytotoxic MeOH and butanol fractions were investigated by in silico molecular docking simulation to assess their binding to HDAC6 and PDK3 and, consequently, their inhibitory effect on HDAC6 and PDK3 as a potential mechanism of the detected cytotoxicity of a lung cancer cell line.

## 2. Results

### 2.1. In-Vitro Cytotoxicity Investigations

Long-term use of current anticancer drugs is associated with serious side effects and a high rate of death. Increased efforts to identify an urgent treatment from natural sources to overcome the rapid rise in lung cancer incidence and the extremely low relative survival rates is one of our research goals. Screening plant extracts cytotoxicity on cancer cell lines is a rapid process for discovering anticancer drugs. Today, many plant-derived anticancer agents are currently in use and/or under clinical trials [18]. *Boswellia serrata* and *Viscum album* extracts, for example, have passed laboratory investigations and are now undergoing clinical trials [19,20].

*C. macrocarpa* (Synonym: *Carissa grandiflora*) is widely cultivated in the KSA as an ornamental plant. It has been evaluated for its cytotoxic activities against MCF-7, NCI-H460, HeLa, and hepatocellular carcinoma (HepG2) [13]. Few flavonoids isolated from the leaves of *C. macrocarpa* have exhibited IC_50_ against the A549 cell line comparable to that of the positive control doxorubicin [21]. It seems that the leaf extract still contains further cytotoxic compounds against the A549 cell line. In this regard, *C. macrocarpa* leaves were extracted using 80% aqueous MeOH, and the obtained extract was fractionated on flash column chromatography using solvents of varied polarity to obtain the *n*-hexane, EtOAc, *n*-butanol, and MeOH fractions (see the Experimental section). The total methanolic extract, as well as its different fractions, were investigated against lung cancer cell line (A549) using the 3-(4,5-dimethythiazol-2yl)-2,5-diphenyltetrazolium bromide (MTT) cell viability assay method. The total extract, as well as the MeOH and butanol fractions, showed dose-dependent cytotoxic activity (Appendix A) with promising IC_50_ values, which are (~8–25 folds) higher than the standard cytotoxic drug, etoposide. The MeOH fraction exerted the most potent cytotoxic activity (IC_50_ = 1.57 ± 0.04 μg/mL), while the total extract exhibited 50% growth inhibition at an almost two-fold concentration (3.3 ± 0.19 μg/mL) as that of the MeOH fraction. The 50% growth inhibitory concentration of the butanol fraction was also almost double (6.16 ± 0.35 μg/mL) that of the total extract. The EtOAc fraction exhibited cytotoxic activity (IC_50_ = 50.66 ± 1.95 μg/mL) comparable to that of the standard drug etoposide (IC_50_ = 50.8 ± 3.16 μg/mL). The *n*-hexane fraction did not produce noticeable cytotoxic effects on the A549 cell line at the maximum examined concentration (100 μg/mL) (Table 1).

These results are evidence that the effect is due to the accumulation of the cytotoxic principles in the polar MeOH fraction, which is, interestingly, the predominant component of the extract (70.9% of the dry extract and 19% of the dry powdered leaves, Table 2).

### 2.2. Antileishmanial Activity

To investigate the anti-*L. major* effects of *C. macrocarpa* total extract and its derived fractions, the promastigotes and axenic amastigotes of *L. major* were separately cultured in the presence and absence of different concentrations (100, 50, 25, 12.5, 6.25, and 3.12 μg/mL). A dose-dependent reduction in the growth of the promastigotes and axenic amastigotes was observed for the EtOAc fraction (IC_50_ = 27.52 ± 0.7 μg/mL, against promastigotes, and 29.33 ± 4.86% growth inhibition of the axenic amastigotes at 100 μg/mL). The butanol fraction exhibited IC_50_ = 73.17 ± 1.62 μg/mL against the promastigotes but negligible growth inhibitory effects against the axenic amastigotes. It is worth noting that *L. major* parasites rotate between attacking mammalian macrophages with intracellular amastigotes and the midgut of sandflies with extracellular promastigotes [22]. In light of this, our research findings on the EtOAc fraction of *C. macrocarpa*, which is being described here for the first time, unquestionably support its usage and development as a dual-function medication that can combat both promastigote and amastigote stages of *L. major*.

### 2.3. Comparative UPLC-ESI-MS/MS Metabolite Profiling of C. macrocarpa Leaves Total Extract and Its Fractions

Altogether 102 metabolites were tentatively identified in the different *C. macrocarpa* leaves total methanolic extract and its polar fractions utilizing the UPLC-ESI-MS/MS in the negative (Appendix A) and positive ionization modes. The compounds are ordered according to their retention time (*R_t_*) in Table 3. The compounds were identified based on their MS and MS^2^ fragmentation data and compared with the literature values. The identified metabolites are classified into 6 major groups (32 organic acids/derivatives, 19 flavonoids/derivatives, 21 flavan-3-ols/derivatives, 10 sterols and triterpenes/derivatives, 12 fatty acids/derivatives, and 8 miscellaneous compounds). Detailed identification comments are shown in the Appendix A.

### 2.4. Molecular Docking

#### 2.4.1. Molecular Docking of Identified Compounds against HDAC6 Enzymes

Targeting HDACs has resulted in a significant increase in non-small-cell lung cancer research over the last decade. Several types of research and preclinical studies have revealed the significant role of HDAC6 inhibitors in the treatment of lung cancer.

The active site of HDAC6 (PDB ID: 5EDU) was recognized by the presence of a co-crystalized ligand trichostatin A. The involved amino acid residues in ligand hydrogen bonding interaction include His 610, His 611, His 651, and Tyr 782, in addition to metal binding with zinc 901 atoms. The molecular simulation of interactions between the identified compounds and the HDAC6 active site was conducted, and the binding affinities pose scores, and binding interactions were studied. The binding affinity values of the compounds with the active site showed high affinities ranging from −23.6583 to −6.8037 kcal/mol.

The binding results revealed that compounds 4″-methyl epigallocatechin gallate, catechin 3-*O*-*β*-D-glucopyranoside, catechin 7-*O*-*β*-D-glucopyranoside, epicatechin 3-*O*-*β*-D-glucopyranoside, and epicatechin-3′-*O*-glucoside showed the best affinity to the HDAC6 active site with pose scores −18.0349, −19.2377, −20.8137, −23.6583, and −17.6491 kcal/mol, respectively (Table 4).

Epicatechin 3-O-*β*-D-glucopyranoside exhibits the highest binding affinity with binding energy −23.6583 kcal/mol (RSMD = 1.70) (Figure 1). The interaction of compounds with the active receptor site is mainly supported by hydrogen bonds and hydrophobic interactions. The interactions are formed by hydrogen bonding with His 610 and His 611 and metal binding with zinc 901 atoms which further form ionic interactions with Asp 649, Asp 742, and His 651, together with hydrophobic interactions with significant distance between the boundary of the active site (Figure 1).

#### 2.4.2. Molecular Docking of Identified Compounds against PDK3

PDK3 is a mitochondrial enzyme that is activated in various human cancers, causing them to progress. Because of its potential therapeutic effect in lung cancer therapy, PDK3 inhibitors have recently been considered a potential target for much research. Considering PDK3, catechin 7-O-*β*-D-glucopyranoside (−21.3350 kcal/mol), hyperoside (−20.6101 kcal/mol), myricetin-3-O-xyloside (−22.7275 kcal/mol), procyanidin B5 (−23.9701 kcal/mol), and procyanidin C2 (−24.2314 kcal/mol) are the top-scoring compounds (Table 4).

Intriguingly, procyanidin C2, which has the best docking binding energy scores −24.2314 kcal/mol (RSMD = 2.23 Å), revealed an interaction with the active site through hydrogen bonding with Lys 134 and Arg 254, with ionic interaction with and Arg 254, π-H interaction with Gly 323 (Figure 2).

## 3. Discussion

Cancer patients are subjected to many therapeutic modalities, including radiotherapy, surgery, immunotherapy, and chemotherapy. The latter is up to now still the major approach used in clinics. However, many patients suffer from chemotherapy-induced side effects [4]. An increasing number of deaths among cancer patients is triggered by chemotherapeutics-related side effects, which necessitates increased efforts to find an urgent solution. In our preceding report, *C. macrocarpa* MeOH fraction was associated with the in vivo protective activity against doxorubicin-induced neurotoxicity [42]. Herein, in our screening of local plants in the KSA to discover anticancer and anti-*L. major* drugs, *C. macrocarpa* leaf extract exhibited potent antineoplastic activity confirmed by the in vitro evaluation of the extract and its fractions against the A549 lung cancer cell line. The results revealed a promising cytotoxic activity for the total extract (IC_50_ = 3.3 ± 0.19) and both of the MeOH (IC_50_ = 1.57 ± 0.04) and butanol (IC_50_ = 6.16 ± 0.35) fractions compared to the standard drug etoposide (IC_50_ = 50.08 ± 3.16). Taken together, *C. macrocarpa* polar fractions, in combination with chemotherapeutic agents, can have chemoprotective and synergistic effects in terms of reducing cancer chemotherapy-associated side effects and enhancing therapeutic efficacy. Metabolite investigation using UPLC-ESI-MS/MS of the *C. macrocarpa* MeOH and butanol fractions revealed the presence of numerous plant polyphenols, particularly the flavonoid and the flavan-3-ol (catechin and epicatechin) derivatives. Flavan-3-ols and flavonols have reportedly been the most extensively researched polyphenols for treating cancer, and many studies have demonstrated their cytotoxic action [43,44]. Therefore, these polyphenolic metabolites would account principally for the cytotoxic effects of *C. macrocarpa* polar fractions. They inhibit tumor development and progression by targeting key signaling transducers, which are controlled in part by epigenetic machinery modulation that includes regulation of the activities of DNA methyltransferases (DNMTs) and HDACs, according to reports [45,46,47]. Furthermore, it has become more apparent how HDAC6 and PDK3 contribute to the development of lung cancer, and it is now generally accepted that the inhibitors may be used to treat lung cancer [16,17]. An in-silico analysis of the binding affinities of variously found polyphenolics against HDAC6 and PDK3 was conducted to shed light on a possible mechanism for in vitro cytotoxic action. Overall, scores of binding affinities of identified ligands suggest that the best binding ligands as inhibitors for HDAC6 mainly belong to epicatechin/catechin glycosides in which the glucosides moiety is essential for interactions. On the other hand, procyanidins are among the highest-scoring compounds as inhibitors for PDK3 in which the phenolic hydroxyl groups and aromatic rings impart in the interaction with the active site residues. The results suggested that polyphenolic compounds of the *C. macrocarpa* polar fractions with elevated pose score could be used as a potential treatment for lung cancer or as a scaffold for developing new inhibitors for enzymes HDAC6 and PDK3, related to lung cancer progression.

On the other hand, the KSA is one of the top ten nations where CL is endemic. Regions of Riyadh, Qassim, Al-Madinah, Al-Hassa, Hail, and Asir recorded the highest prevalence [8]. Under these circumstances, which are exacerbated by the absence of a prophylactic vaccine and/or safe, affordable treatment, an in vitro anti-*L. major* assay of leaf extracts of *C. macrocarpa* against promastigotes and axenic amastigotes was performed in this research. Particularly, the EtOAc fraction exhibited noticeable growth inhibitory effects against the promastigotes and axenic amastigotes, and to a lesser extent, the butanol fraction inhibited the growth of the promastigotes form of the parasite.

Metabolites investigation of the EtOAc fraction led to the identification of 40 compounds, mainly hydroxycinnamoyl, triterpene, flavanol, flavonoid, sterols, and fatty acids derivatives. Reviewing the literature revealed that most of these metabolites exhibit antileishmanial activity against amastigotes and/or promastigotes forms of one or more *Leishmania* spp.

Among these metabolites, the hydroxycinnamoyl derivatives caffeic and ferulic acids were reported to inhibit promastigotes and intracellular amastigotes of *L. amazonensis* [48,49]. Oleuropein, a phenyl propanoid, inhibited promastigotes growth of *L. major* [50] and *L. donovani* [51].

A combination of oleanolic and ursolic acids has been shown to have a powerful growth-inhibitory impact against *L. amazonensis* and *L. braziliensis* amastigote form [52]. Additionally, in a mouse model of CL brought on by *L. amazonensis*, an extract from the leaves of *Baccharis uncinella* rich in oleanolic and ursolic acids has shown a leishmanicidal effect [53]. Ursolic acid showed more action against *L. amazonensis* promastigotes than oleanolic acid when each compound was examined independently. Oleanolic acid, but not ursolic acid, could kill amastigotes by causing macrophages to produce nitric oxide. Using BALB/c mice infected with *L. amazonensis*, this capability was also demonstrated in vivo [53]. Ursolic acid, obtained from the leaves of the *B. uncinella* plant, has also demonstrated potent effects against experimental visceral leishmaniasis brought on by *L. infantum*, and it has been found to reduce the parasite load in the spleen and liver [54]. Additionally, for the treatment of visceral leishmaniasis brought on by *L. donovani*, ursolic acid in a delivery system made from a nanostructured lipid carrier coated with an N-octyl-chitosan surface was more efficient than free ursolic acid [55,56]. Betulinic acid has recently been introduced as an antileishmanial compound, and betulin heterocyclic derivatives, including betulinic acid, have antiparasitic activity against *L. donovani*. Its molecular mechanism was suggested to be the induction of apoptosis through the inhibition of DNA topoisomerase I and II activity [57].

Gallocatechin and epigallocatechin gallate, two derivatives of the flavan-3-ol, significantly reduced the proliferation of *L. amazonensis* promastigotes while having little to no cytotoxicity on murine macrophages and human RBCs [58]. Additionally, topical treatment with epigallocatechin gallate significantly decreased the lesion size and parasite load of the intracellular amastigotes of *L. amazonensis* [59]. It has been demonstrated that the anti-leishmanial effects of catechin, epicatechin, epicatechin gallate, and epigallocatechin-3-gallate are mediated by the host’s immune response to the parasite’s defense [60].

Gallic acid, a phenolic acid, has been discovered to be effective against *L. major* and *L. donovani promastigotes* [9,61], and it has also shown promise as an adjuvant to standard amphotericin B for the treatment of CL [62].

The flavonoid quercetin has been shown to inhibit the growth of the promastigotes and intracellular amastigotes of *L. amazonensis* [49]. Additionally, quercetin has been shown useful in treating mice with leishmaniasis. It causes *L. major* promastigotes to undergo caspases-independent apoptosis, and it triggers the demise of infected BALB/c mouse phagocytes [63].

Experiments on the role of fatty acids in leishmaniasis treatment found that linoleic acid suppresses the release of *L. donovani* macrovesicles and enhances the Th-1 type cytokines immune response [64]. Additionally, it reduces the survival of microvesicles generated from *L. donovani* in macrophages and prevents their release [64].

After reading up on the antileishmanial properties of stigmasterol and β-sitosterol, two common plant metabolites found in most plants and detected in the EtOAc fraction, it was found that these two compounds effectively damaged promastigote and amastigote forms of *Leishmania major*, *L. amazonensis*, *L. tropica*, and *L. donovani* [65,66,67].

Taken together, the data make it abundantly evident that *C. macrocarpa* EtOAc fraction merits preclinical animal testing to be developed as an anti-*L. major* medication.

## 4. Materials and Methods

### 4.1. General Experimental Procedures

The lung adenocarcinoma cell line (A549) was obtained from the Japanese Collection of Research Bioresources (JCRB) Cell Bank, National Institute of Biomedical Innovation. Institute of Tropical Medicine, Nagasaki University supplied us with *L. major*, which was used as endorsed by the ethical board of the Institute of Tropical Medicine, Nagasaki University, Japan. M199 and Dulbecco’s Modified Eagle’s Media, kanamycin, dimethyl sulfoxide (DMSO), etoposide, miltefosine, fetal bovine serum, and MTT were obtained from Nacalai Tesque in Kyoto, Japan. The 96-well plates were purchased from Becton Dickinson (Franklin Lakes, NJ, USA). Solvents for the extraction, fractionation, and mass analysis were obtained from Sigma-Aldrich Inc., St. Louis, MO, USA.

### 4.2. Plant Material

During the blossoming time (October 2019), *C. macrocarpa* leaves were collected from a tree (accessed on 15 October 2020) near the Faculty of Engineer, Najran University, KSA (https://maps.app.goo.gl/4vnxmEuyqaWGuktc6, accessed on 15 October 2020). A verifier sample (CM 1019) was deposited in the Pharmacognosy Department, College of Pharmacy, Najran University.

### 4.3. Extraction and Fractionation

The total hydroalcoholic extract of *C. macrocarpa* leaves (250 g) was obtained using homogenization in MeOH–H_2_O (8:2, *v*/*v*, 4 × 1.5 L) at ambient temperature. The extract was dried under reduced pressure at 40 °C. A total of 56 g of the dry extract (66.6 g) was fractionated using a flash chromatography column (15 × 7 cm, i.d.) packed with silica gel (70–230 mesh, Merck, Darmstadt, Germany) and operated using *n*-hexane, EtOAc, *n*–butanol, and MeOH (3 L each), successively. Vacuum drying of the different eluates afforded the dry fraction weights listed in Table 2.

### 4.4. Pharmacological Investigation

#### 4.4.1. Preparation of the Stock Solutions and the Serial Concentrations

To make the initial stock solution (10 mg/mL) of the plant samples, 10 mg of each of the total extract and the fractions were dissolved in 1 mL of DMSO. The second stock solution (5 mg/mL) was generated by mixing 0.5 mL of this stock solution with 0.5 mL of DMSO, which was mixed well. We continued two-fold dilution until we reached the final stock solution (0.312 mg/mL). To make the final serial concentrations, 1 μL aliquots of the sample stock solutions in two-fold dilution (10, 5, 2.5, 1.25, 0.625, and 0.312 mg/mL) were added to the corresponding well (the final volume became 100 μL) to obtain the final concentrations (100, 50, 25, 12.5, 6.25, and 3.12 μg/mL).

#### 4.4.2. Cytotoxicity Investigation

The colorimetric cell viability MTT assay was used to determine the cytotoxicity against the A549 cell line based on the published procedures [9]. The cells were cultured with 10% heat-inactivated fetal calf serum, kanamycin (100 μg/mL), and amphotericin B (5.6 μg/mL) in Dulbecco’s Modified Eagle Medium. A total of 1 μL aliquots of the previously prepared sample stock solutions and 99 μL media containing 5 × 10^3^ lung cancer cells were applied to each well in a 96-well plate, except 3 wells were not inoculated with cells to serve as a blank. The plate was then incubated at 37 °C under a 5% CO_2_ atmosphere for 72 h. The medium was aspirated, and a 100 μL solution of 0.5 mg/mL MTT was added to each well, and the plate was further incubated for 1.5 h. The formazan crystals, the MTT reduction product, were dissolved in DMSO (100 μL/well). The absorption of each well at 540 nm was measured using the Molecular Devices Versamax Tunable Microplate Reader. DMSO was used as a negative control, and etoposide as a positive control. A total of 1 μL of each etoposide stock solution (5–0.019 mg/mL, two-fold dilution) was added to the corresponding wells of a 96 microplate, resulting in final serial concentrations of (50, 25, 12.5, 6.25, 3.12, 1.56, 0.78, 0.39, and 0.19 μg/mL). All data have been reported as the mean ± SE of triplicate results. The cytotoxic activity was calculated from the equation:

% inhibition = [1 − (*A*_test_ − *A*_blank_)/(*A*_control_ − *A*_blank_)] × 100. Where *A*_control_ is the absorbance of the control (DMSO) well, *A*_test_ is the absorbance of the test wells, and *A*_blank_ is the absorbance of the cell-free wells [9].

#### 4.4.3. Antileishmanial Promastigotes and Axenic Amastigotes Assays

The colorimetric cell viability MTT test was used to measure the growth-inhibitory activity of *L. major* promastigotes and axenic amastigotes. In the log growth phase, the promastigotes were cultured in an M199 medium supplemented with 10% heat-inactivated fetal bovine serum and 100 μg/mL kanamycin [9]. Axenic amastigotes of *L. major* were prepared in a cell-free medium at pH 5.5 and 37 °C according to the previously described method [68,69]. The medium containing the *L. major* cells was transferred to a Falcon 50 mL sterilized conical centrifuge tube and centrifuged for 15 min at 5000 rpm until the cells were completely settled, the old medium was aspirated, and 5 mL of fresh medium was added to the settled cells, which were counted under the microscope and then diluted to a concentration of (1 × 10^5^ cells/99 μL medium). In a 96-well plate, 99 μL medium containing 1 × 10^5^ of *L. major* cells and 1 μL aliquots (the final volume became 100 μL) of the sample stock solutions to make final concentrations (100, 50, 25, 12.5, 6.25, and 3.12 μg/mL) was added to each well, except 3 wells that were left free of cells to serve as a blank. The plate was incubated under an ambient atmosphere for 72 h. The plate was then centrifuged for 5 min at 200 G, and the medium from each well was aspirated. A solution (100 μL) of MTT (0.5 μg/mL) was added, and the incubation was continued for another 12 h. The produced formazan crystals from the MTT reduction were then dissolved in DMSO [9]. The absorbance of each well was measured at 540 nm using a Molecular Devices Versamax tunable microplate reader. Miltefosine was used as a positive control. All data have been reported as the mean ± SE of triplicate results [9].

The viability of axenic amastigote was evaluated by the intensity of MTT deposition by Image J software v1.47 (National Institutes of Health, Bethesda, MD, USA)[70]. The inhibitory activity was calculated as follows: % inhibition = [1 − (*A*_test_ − *A*_blank_)/(*A*_control_ − *A*_blank_)] × 100. Where *A*_contro_l is the intensity of the negative control (DMSO) well, *A*_test_ is the intensity of the test wells, and *A*_blank_ is the intensity of the cell-free wells [9].

### 4.5. UPLC-ESI-MS/MS Investigation

Solutions (100 μg/mL) from the total extract and its fractions (*Viz*. EtOAc, *n*-butanol, and MeOH) were prepared using analytical grade MeOH, then filtered (a 0.2 μm membrane disc filter) to be ready for UPLC-ESI-MS/MS analysis as follows: ESI-MS positive and negative ion acquisition mode was carried out on a XEVO TQD triple quadruple instrument (Waters Corporation, Milford, MA01757 U.S.A) mass spectrometer. The separation was achieved using an ACQUITY UPLC-BEH C18 1.7–2.1 µm × 50 mm column, flow rate (0.2 mL/min). Gradients of the solvent system 0.1% formic acid in H_2_O (A)/0.1 %formic acid in MeOH (B) were applied in the following schedule: A/B, 9:1 (5 min), 7:3 (10 min), 3:7 (7 min), 1:9 (4 min), 0:10 (3 min), and 9:1 (3 min). The samples were injected automatically using a Waters ACQUITY FTN autosampler. The instrument was controlled by MassLynx 4.1 software (Water Corporation, Milford, Massachusetts, United States). The MS operated in the negative mode with a capillary voltage of 30 eV, 3 kV; desolvation temperature, 450 °C; cone gas flow, 50 L/h; and desolvation gas flow, 900 L/h. A source temperature of 150 °C and high purity nitrogen as a sheath and auxiliary gas at a flow rate of 80 and 40 (arbitrary units), respectively. A collision energy of 35% was used in MS/MS fragmentation. Mass spectra were detected in the ESI negative ion and positive ion modes between *m/z* 50 and 2000 [71].

### 4.6. Molecular Docking

Docking analysis was conducted using Molecular Operating Environment software (MOE 2014.0901). The crystal structures of HDAC6 (PDB ID: 5EDU) [17] and PDK3 (PDB ID: 1Y8O) [16] were obtained from the Protein Data Bank (https:/www.rscb.org/pdb) [72]. The protein receptor structures were prepared and optimized for docking using the MOE Ligx option. The receptor’s active sites for ligand binding were determined based on amino acid residues interacting with the complexed ligand for each protein. Unessential residues and water molecules were removed. The compounds summarized in Table 4 and illustrated in Appendix A were imported to MOE and subjected to energy minimization using MMFF94x force field, and a virtual ligand database was constructed. The molecular docking simulation was performed through flexible ligand-fixed receptor docking using Triangle Matcher placement, forcefield refinement with London dG as scoring, and rescoring algorithm. The docking score, root mean square deviation (RSMD), and 2D and 3D interactions were recorded.

## 5. Conclusions

The MeOH fraction obtained from *C. macrocarpa* leaf extract exhibited promising cytotoxic activity against the A549 cell line. The results may be attributed to polyphenolics (epicatechin/catechin glycosides, procyanidins, flavonoids, and phenolic acid derivatives) detected in the fraction by the LC-MS/MS analysis. The MeOH fraction could be developed as a new therapeutic alternative to chemotherapies in treating lung cancer and ameliorating chemotherapeutics-induced side effects. Our research also provides computational evidence that these flavan-3-ol derivatives may target PDK3 and HDAC6 in the suppression of lung malignancy. The *L. major* promastigotes and axenic amastigotes were susceptible to putative growth inhibitory effects from the EtOAc fraction. Even though there is currently no evidence to support the anti-*L. major* from in vivo animal models, the LC-MS/MS examination clearly showed the existence of several metabolites with well-established in vivo antiprotozoal action against various *Leishmania* spp. The EtOAc fraction metabolites of *C. macrocarpa* may serve as a scaffold for the development of anti-*L. major* therapy and/or as a source of antileishmanial metabolites.

## Figures and Tables

**Figure 1 pharmaceuticals-15-01561-f001:**
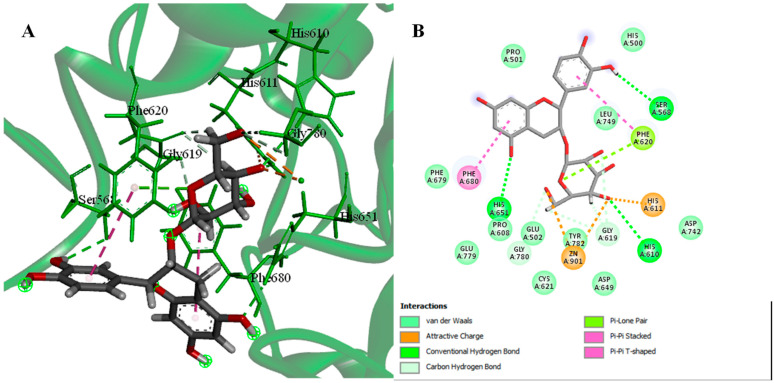
3D (**A**) and 2D (**B**) interactions complex of epicatechin-3-*O*-glucopyranoside with HDAC*6* (PDB ID: 5EDU).

**Figure 2 pharmaceuticals-15-01561-f002:**
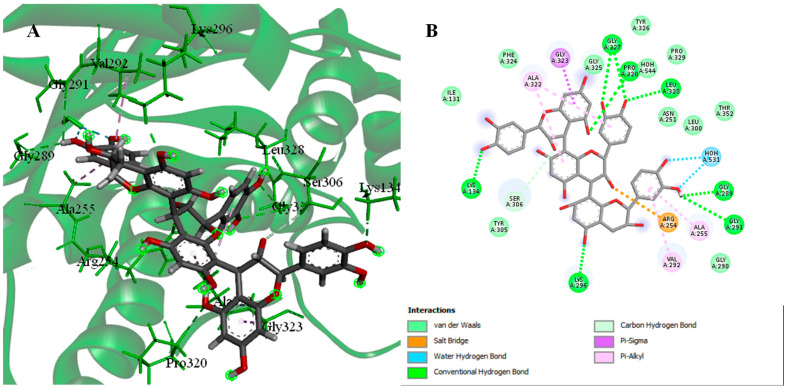
3D (**A**) and 2D (**B**) interactions complex of procyanidin C2 with PDK3 (PDB ID: 1Y8O).

**Table 1 pharmaceuticals-15-01561-t001:** Lung adenocarcinoma cytotoxicity and anti-*L. major* activity of *C. macrocarpa* leaves.

	Cytotoxicity	Anti-*L. major* Activity			
Sample	A549 Cell Line	Promastigotes		Axenic Amastigotes	
	IC_50_ ± SE (μg/mL)	% Inhibition ± SE (at 100 μg/mL)	IC_50_ ± SE (μg/mL)	% Inhibition ± SE (at 100 μg/mL)	IC_50_ ± SE (μg/mL)
Total ext.	3.3 ± 0.19	33.21 ± 2.39	>100	7.42 ± 1.4	>100
MeOH fr.	1.57 ± 0.04	nil	>100	nil	>100
Butanol Fr.	6.16 ± 0.35	61.15 ± 0.86	73.17 ± 1.62	nil	>100
EtOAc fr.	50.66 ± 1.95	88.59 ± 1.68	27.52 ± 0.7	29.33 ± 4.86	>100
Hexane fr.	>100	nil	>100	nil	>100
Etoposide	50.08 ± 3.16	-	-	-	-
Sodium stibogluconate	-	-	-	100	22.45 ± 2.22
Miltefosine	-	100	6.39 ± 0.29	100	12.35 ± 1.8

nil: Negligible growth inhibitory effects. SE: Standard error of triplicate experiments

**Table 2 pharmaceuticals-15-01561-t002:** Weights of total extract and its flash chromatography obtained fractions and their percentage in the dry powdered *C. macrocarpa* leaves.

Sample	Weight (g)	% In the Total Extract	% In the Dry Powder
Total ext.	66.6	100%	26.64%
MeOH fr.	47.2	70.87%	18.88%
Butanol Fr.	3.63	5.45%	1.452%
EtOAc fr.	5.52	8.29%	2.208%
Hexane fr.	0.2	0.3%	0.08%

**Table 3 pharmaceuticals-15-01561-t003:** Secondary metabolites identified in the *C. macrocarpa* total extract and its fractions.

Peak No.	*R_t_*	Molecular Weight	MS [M − H]^−^/[M + H]^+^	MS^2^	Tentatively Identified Compound	Reference	Class	Total	EtOAc fr.	Butanol fr.	MeOH fr.
**1**	0.74	378	377.1617/379.2518	333, 271, 257, 163, 119	Carinol	[15]	Miscellaneous	√	-	√	√
**2**	0.75	192	191.0583/193.1359	173	Quinic acid	[23]	Phenolic acid	-	√	-	-
**3**	0.80	234	233.1027/-	214, 164, 134	Dehydrocarissone (11-hydroxy-1,4-eudesmadien-3-one)	[14,15]	Miscellaneous	-	√	-	-
**4**	0.85	240	239.0533/-	221, 203, 188, 173, 143	Cryptomeridiol	[14,15]	Miscellaneous	-	-	√	-
**5**	0.96	418	-/419.2318	386, 359, 356, 255	3′-(4″-methoxyphenyl)-3′-oxo-propionyl hexadecanoate	[15]	Fatty acid	√	-	-	-
**6**	1.87	354	353.1642/355.1283	191,179,161	3-*O*-Caffeolyquinic acid	[11]	Phenolic acid	√	-	-	√
**7**	2.03	354	353.1728/-	191,173,161	4-*O*-Caffeolyquinic acid	[11]	Phenolic acid	-	-	-	√
**8**	2.13	326	325.1349/-	187, 163, 145	Coumaroyl-β-glucose	[24,25]	Phenolic acid	-	-	√	√
**9**	2.161	354	353.2025/-	191,179,161	*5*-*O*-Caffeolyquinic acid	[11]	Phenolic acid	-	-	√	√
**10**	2.24	578	577.2717/-	425, 289	Type B (epi)catechin dimer	[11]	Flavan-3-ol	√	-	√	√
**11**	2.45	866	865.4962/-	451, 425, 407, 289	Type B (epi)catechin trimer	[11]	Flavan-3-ol	√	-	√	√
**12**	2.85	320	319.1711/-	301, 275, 257, 231, 203,163, 119	5-*O*-*p*-Coumaroylshikimic acid	[24]	Phenolic acid	√	-	√	√
**13**	2.87	342	-/343.1820	326, 311, 285	Caffeic acid 3-glucoside	[26]	Phenolic acid	√	-	√	√
**14**	2.87	452	451.3018/-	408, 393, 351, 337, 301, 273, 245	Catechin-3-*O*-glucoside	[27]	Flavan-3-ol	-	-	-	√
**15**	2.87	320	319.1711/-	275, 257, 199, 163, 119	4-*O*-*p*-Coumaroylshikimic acid	[24]	Phenolic acid	√	-	√	√
**16**	2.87	452	451.3018/-	391, 343, 301, 287, 273, 247	Epicatechin-3-*O*-glucoside	[27]	Flavan-3-ol	-	-	-	√
**17**	3.00	290	289.0894/-	245, 205, 203, 187, 179, 161	(epi) Catechin	[27]	Flavan-3-ol	√	-	√	√
**18**	5.05	756	755.4382/-	593, 285	Kaempferol-7*-O*-hexoside-3-*O*-rutinoside	[11]	Flavonoid	√	-	√	√
**19**	5.12	756	755.5181/-	609, 301	Quercetin-7*-O*-deoxyhexoside-3-*O*-deoxyhexosyl-hexoside	[11]	Flavonoid	√	-	√	√
**20**	5.14	452	451.3743/-	391, 343, 301, 287, 273, 247	Epicatechin-3-*O*-glucoside isomer	[27]	Flavan-3-ol	√	-	√	√
**21**	5.52	740	739.3967/-	593, 285	Kaempferol-7*-O*-deoxyhexoside-3*-O*-deoxyhexosyl-hexoside isomer 1	[11]	Flavonoid	√	-	√	√
**22**	5.59	740	739.4193/-	593, 285	Kaempferol-7*-O*-deoxyhexoside-3*-O*-deoxyhexosyl-hexoside isomer 2	[11]	Flavonoid	-	-	-	√
**23**	5.59	610	609.2661/-	301	Quercetin- 3*-O*-deoxyhexosyl-hexoside isomer 1	[11]	Flavonoid	-	-	-	√
**24**	5.75	610	609.3278/611.2927	465, 303	Quercetin- 3*-O*-deoxyhexosyl-hexoside isomer 2	[11]	Flavonoid	√	-	√	√
**25**	5.80	578	577.2878/-	425, 289	Type B (epi)catechin dimer	[11]	Flavan-3-ol	-	-	√	√
**26**	5.90	450	449.1666/-	317, 316	Myricetin-3-O-xyloside	[23,28]	Flavonoid	-	-	√	√
**27**	6.04	300	-/302.8930	275, 257, 229, 215, 153	Quercetin	[23]	Flavonoid	-	-	√	√
**28**	6.13	594	593.3281/-	557, 467, 441, 425, 407, 289	(epi) Gallocatechin-(epi)catechin	[28]	Flavan-3-ol	-	-	√	√
**29**	6.58	516	515.2966/-	353, 179	Dicaffeoylquinic acid	[24]	Phenolic acid	√	-	√	√
**30**	6.80	138	136.9441/-	109, 93	Hydroxy benzoic acid	[23]	Phenolic acid	-	-	√	√
**31**	6.87	188	187.1353/-	169, 125	Gallic acid monohydrate	[23]	Phenolic acid	-	√	-	-
**32**	6.87	194	193.1353/-	169, 125	Ferulic acid	[23]	Phenolic acid	-	√	-	-
**33**	7.01	594	593.4289/-	557, 467, 441, 425, 407, 289	(epi) Gallocatechin-(epi)catechin	[28]	Flavan-3-ol	-	-	√	-
**34**	8.11	180	178.8018/-	179, 135	Caffeic acid	[23]	Phenolic acid	-	√	√	-
**35**	8.12	198	196.9333/-	120, 104, 93, 87	Syringic acid	[23]	Phenolic acid	-	-	-	√
**36**	8.71	328	327.2579/-	281, 279, 255, 213, 183	Trihydroxy-octadecadienoic acid	[29]	Fatty acid	√	√	-	-
**37**	9.14	574	573.6962/-	397, 223, 173	Feruloyl-*O*-sinapoylquinic acid	[24]	Phenolic acid	√	-	-	√
**38**	9.30	330	329.2814/-	311, 293, 229, 211, 171, 143	Trihydroxy-octadecenoic acid	[29]	Fatty acid	√	-	-	-
**39**	9.94	940	939.0552/-	778, 735, 732, 717, 571	Diacetoxy-5-methoxyphenyl acroyl-*O*-*p*-coumaroyl-*O*-caffeoylquinic acid derivative	[23]	Phenolic acid	√	-	√	√
**40**	10.21	310	309.2307/-	291, 279, 251, 223, 221, 89	Dihydroxy-octadecadienoic acid	[29]	Fatty acid	-	√	-	-
**41**	10.21	378	377.1844/-	345, 327	Oleuropein aglycone	[15]	Miscellaneous	-	√	-	-
**42**	10.22	342	341.2001/-	326, 311, 285	Tetramethoxyflavone	[30]	Flavonoid	-	-	√	-
**43**	10.29	310	309.2261/-	291, 279, 251, 223, 221, 89	Dihydroxy-octadecatrienoic acid isomer	[29]	Fatty acid	-	√	-	-
**44**	11.21	344	343.2618/-	191, 169	Galloylquinic acid	[23]	Phenolic acid	-	-	√	-
**45**	11.40	306	305.1978/-	261, 219, 221, 179, 165, 125	(epi)-Gallocatechin	[28]	Flavan-3-ol	-	√	-	-
**46**	11.54	294	293.1970/-	275, 251, 221, 207	Monohydroxy-Octadecatrienoic acid	[29]	Fatty acid	-	√	-	-
**47**	12.06	176	174.9668/-	157	Ascorbic acid	[23]	Phenolic acid	-	-	√	-
**48**	12.60	288	287.3836/-	151, 135, 125, 107	Eriodictyol	[30]	Flavonoid	-	-	√	-
**49**	12.78	436	-/437.2773	307, 181	Epigallocatechin-3-*O*-cinnamate	[31]	Flavan-3-ol	-	-	√	-
**50**	12.98	472	471.4764/-	441, 407, 313, 303, 287, 269, 257, 243, 161,	Methyl-3-*O*-gallocatechin gallate	[27]	Flavan-3-ol	-	√	√	-
**51**	12.98	454	-/455.3868	439, 411, 393, 248, 207, 203, 191, 189	3*β*-hydroxyolean-11-en-28,13*β*-olide	[14,32]	Triterpene	-	√	-	-
**52**	13.11	414	413.3687/-	366, 270, 255, 189, 175, 161	*β*-Sitosterol	[33,34]	Sterol	-	√	-	-
**53**	13.26	578	577.4777/-	425, 289	(epi) Catechin dimer	[11]	Flavan-3-ol	-	-	-	√
**54**	13.30	544	543.5621/-	353, 173	Dimethoxycinnamoyl-*O*-caffeoylquinic acid	[23]	Phenolic acid	-	-	-	√
**55**	13.35	312	311.2344/-	293, 275, 253, 235, 223	Dihydroxy-octadecadienoic acid	[29]	Fatty acid	-	√	-	-
**56**	13.78	472	471.4489/-	441, 407, 303, 288, 257, 243, 201, 169, 161	Methyl-3-*O*- epigallocatechin gallate	[27]	Flavan-3-ol	-	√	√	-
**57**	14.37	634	633.4683/-	481, 305	(epi)Gallocatechin-*O*-gallate-O-glucuronide	[27]	Flavan-3-ol	-	-	√	-
**58**	14.55	432	431.1743/-	341, 311	Vitexin	[23]	Flavonoid	-	√	-	-
**59**	14.69	560	559.4719/-	397, 223	3-*O* Caffeoyl-4-*O*-sinapoylquinic acid	[24]	Phenolic acid	-	-	√	-
**60**	14.79	560	559.6325/-	397, 223	3-*O*-Sinapoyl-4-*O*-caffeoylquinic acid	[24]	Phenolic acid	-	-	√	-
**61**	14.82	472	471.4497/-	441, 407, 297, 269, 241, 213, 199, 168, 161	Methyl-3-*O*-epigallocatechin gallate isomer	[27]	Flavan-3-ol	-	√	√	-
**62**	15.03	472	471.4144/-	453, 435, 407, 389	23-Hydroxybetulinic acid	[14]	Sterol	-	√	-	-
**63**	15.06	560	559.4719/-	397, 223	1-*O*-Caffeoyl-3-*O*-sinapoylquinic acid	[24]	Phenolic acid	-	-	√	-
**64**	15.06	354	353.3186/-	179,161	4-*O*-Caffeoylshikimic acid	[24]	Phenolic acid	-	-	√	-
**65**	15.30	618	617.6021/-	599, 465, 289	(epi) Catechin-*O*-gallate-*O*-glucuronide	[35]	Flavan-3-ol	√	-	√	-
**66**	15.28	296	295.2867/	277, 253, 223, 167	13-hydroxyoctadec-2-enoic acid	[29]	Fatty acid	-	√	-	-
**67**	15.30	470	-/471.3879	456, 439, 411, 393, 248, 207, 203, 191, 189	Methyloleanolate	[14]	Triterpene	-	√	-	-
**68**	15.46	648	647.6069/-	485, 470, 455, 440	3,27-Dihydroxy-12-ursen-28-oic acid; 3*β*-form, 27-(4-Hydroxy-3-methoxycinnamoyl) (*E*-form)	[14]	Triterpene	-	√	√	-
**69**	15.48	454	-/455.4161	307, 179, 137	Gallocatechin derivative	[36]	Flavan-3-ol	-	√	√	-
**70**	15.66	544	543.3333/-	353, 173	Dimethoxycinnamoyl-*O*-caffeoylquinic acid isomer	[23]	Phenolic acid	-	-	-	√
**71**	15.70	618	617.5588/-	599, 465, 289	(epi)Catechin-*O*-gallate-*O*-glucuronide isomer	[35]	Flavan-3-ol	-	-	√	-
**72**	15.70	454	-/455.4180	307, 179, 137	Gallocatechin derivative	[36]	Flavan-3-ol	-	√	√	-
**73**	15.92	646	647.5842/-	485, 470, 455, 440	27-Coumaroyloxyursolic acid	[23]	Triterpene	-	√	-	-
**74**	16.52	354	352.9919/-	179,161	3-*O*-Caffeoylshikimic acid	[24]	Phenolic acid	-	-	-	√
**75**	16.59	382	-/383.2513	369, 351, 195	Dimethoxycinnamoylquinic acid	[24]	Phenolic acid	√	-	√	√
**76**	16.59	326	325.1193/-	187, 163, 145	Coumaric acid hexoside	[25]	Phenolic acid	-	-	√	-
**77**	17.08	338	337.1823/-	202, 190, 163	*p*-Coumaroylquinic acid	[23]	Phenolic acid	-	-	√	-
**78**	17.55	382	381.2726/-	367, 349, 193	Dimethoxycinnamoylquinic acid	[24]	Phenolic acid	-	-	√	-
**79**	18.45	600	-/601.5302	447, 313, 285, 284, 169, 151, 125	Kaempferol galloylglucoside	[37]	Flavonoid	√	-	-	-
**80**	18.58	456	455.4745/-	439, 419, 411, 410, 407, 397	Ursolic acid	[28,32]	Triterpene	√	√	-	-
**81**	19.06	456	455.4576/-	439, 419, 411, 410, 407, 397	Carissic acid (isomer of ursolic acid)	[28,32]	Triterpene	√	√	√	-
**82**	19.16	456	455.4955/457.4337	439, 419, 411, 410, 407, 397	Oleanolic acid	[28,32]	Triterpene	√	√	√	-
**83**	20.28	340	339.3065/-	295, 251, 179	Caffeoyl-2-hydroxyethane-1,1,2-tricarboxylic acid	[23]	Phenolic acid	-	√	√	-
**84**	20.73	376	375.3291/-	361, 347, 294, 123	Carissanol	[15]	Miscellaneous	-	-	√	-
**85**	20.81	238	-/239.2097	221	Germacrenone	[14,15]	Miscellaneous	√	-	√	-
**86**	20.92	280	279.2659/-	237, 222, 208, 194, 166, 152, 137, 111, 97, 83, 69, 57, 43	Linoleic acid	[32]	Fatty acid	-	√	-	-
**87**	21.35	318	316.9698/-	299, 289, 273, 245	Dimethyl (epi)catechin	[27]	Flavan-3-ol	-	√	-	-
**88**	21.55	594	593.4359/-	285	Kaempferol-*O*-deoxyhexosyl-hexoside	[37]	Flavonoid	√	-	√	-
**89**	22.09	594	593.4612/-	285	Kaempferol-3-*O*-rutinoside	[37]	Flavonoid	√	-	√	-
**90**	22.29	328	327.4435/-	309, 239, 229, 211, 171, 163	Oxo-dihydroxy-octadecenoic acid	[29]	Fatty acid	-	√	-	-
**91**	22.36	328	327.4144/-	309, 239, 229, 211, 171, 163	Oxo-dihydroxy-octadecenoic acid isomer	[29]	Fatty acid	-	√	-	-
**92**	22.51	256	255.2160/-	211, 183, 155, 127, 99	Palmitic acid	[32]	Fatty acid	-	√	-	-
**93**	22.78	440	-/441.3986	323,179, 161, 133	Caffeoyl cyclohexanediol hexoside	[38]	Phenolic acid	-	√	-	-
**94**	22.90	422	-/423.4181	307, 163, 145, 119	*p*-Coumaroyl cyclohexanediol hexoside	[38]	Phenolic acids	-	√	-	-
**95**	23.12	328	317.0337/-	179, 151, 137	Myricetin	[39]	Flavonoid	√	√	-	-
**96**	25.30	412	-/413.3087	395, 256, 214	Stigmasterol	[33,34]	Sterol	√	√	√	-
**97**	25.85	318	316.9427/-	299	Methyldihydroquercetin (Cedeodarin)	[39]	Flavonoid	√	√	-	-
**98**	26.53	612	-/613.6197	595, 521, 491, 449, 327, 287	Rhamnosyl-hexosyl-methyl-quercetin	[38]	Flavonoid	-	-	√	-
**99**	27.02	464	-/465.4451	301, 300, 257, 255, 229, 179. 151	Hyperoside	[38]	Flavonoid	-	-	√	√
**100**	27.21	464	-/465.4246	301, 300, 257, 255, 229, 179. 151	Isoquercetin	[38]	Flavonoid	-	-	√	√
**101**	27.31	622	621.6783/-	501	2(R)-26-([(2*E*)-3-(4-hydroxy-3-methoxyphenyl)-1-oxo-2- propen-1-yloxy)-2,3-dihydroxypropyl ester	[40]	Miscellaneous	√	-	√	√
**102**	31.25	430	429. 3132/430.9172	205, 191, 177, 149, 121	*α*-Tocopherol	[41]	Miscellaneous	√	√	√	√

**Table 4 pharmaceuticals-15-01561-t004:** Docking score of detected polyphenolics and their isomers against HDAC6 and PDK3 enzymes.

No.	Compound Name	HDAC6		PDK3	
		Score (kcal/mol)	RMSD Refine (Å)	Score (kcal/mol)	RMSD Refine (Å)
**6**	3-*O*-Caffeolyquinic acid	−10.7626	1.40	−13.1642	1.59
**7**	4-*O*-Caffeolyquinic acid	−16.1666	1.09	−17.3380	1.26
**8**	Coumaroyl-5-*β*-glucose	−9.1090	1.22	−14.4638	0.97
**9**	5-*O*-Caffeolyquinic acid	−11.5285	1.77	−14.4531	1.44
**10**	Procyanidin Bl	−8.8345	2.19	−15.8844	1.85
**10a**	Procyanidin B2	−9.5399	1.93	−17.5420	1.73
**10b**	Procyanidin B3	−10.7608	1.26	−17.4908	1.81
**10c**	Procyanidin B4	−13.3266	2.04	−16.6954	1.55
**10d**	Procyanidin B5	−9.8460	2.31	−23.9701	1.45
**10e**	Procyanidin B6	−12.0986	1.64	−19.2123	1.27
**10f**	Procyanidin B8	−9.9105	1.38	−18.6158	2.19
**11**	Procyanidin C2	−10.8164	2.61	−24.2314	2.23
**13**	Caffeic acid 3-glucoside	−17.0231	1.23	−17.1953	1.15
**14**	Catechin 3-*O*-*β*-D-glucopyranoside	−19.2377	2.56	−15.7921	1.40
**14a**	Catechin 5-*O*-*β*-D-glucopyranoside	−15.7346	1.57	−15.8820	1.62
**14b**	Catechin 7-*O*-*β*-D-glucopyranoside	−20.8137	1.38	−21.3350	1.86
**15**	4-*O*-*p*-Coumaroylshikimic acid	−14.0624	0.89	−14.3744	1.29
**16**	Epicatechin 3-*O*-*β*-D-glucopyranoside	−23.6583	1.70	−19.5019	1.39
**16a**	Epicatechin 6-C-glucoside	−15.2686	1.80	−14.3566	1.26
**16b**	Epicatechin 8-C-glucoside	−7.8402	2.12	−17.8381	1.60
**16c**	Epicatechin-3′-*O*-glucoside	−17.6491	1.42	−14.1809	1.87
**17**	Catechin	−10.1261	0.93	−15.4482	0.83
**17a**	Epicatechin	−7.3314	1.64	−14.7507	0.97
**26**	Myricetin-3-*O*-xyloside	−9.5913	1.23	−22.7275	1.36
**27**	Quercetin	−12.8749	0.86	−14.3401	1.29
**29**	1,4-Dicaffeoylquinic acid	−12.3875	1.16	−18.8016	1.35
**33**	Epigallocatechin-(4-*β*-6)-(+)-catechin	−11.4416	2.09	−16.0503	1.89
**33a**	Epigallocatechin-(4-*β*-8)-catechin	−9.4011	1.35	−19.7638	1.97
**33b**	Epicatechin (4 *β*.8) epigallocatechin	−9.1194	1.82	−19.5053	1.61
**33c**	Catechin-(4*α*-8)-(-)-epigallocatechin	−12.9583	1.26	−18.4410	1.08
**35**	Syringic acid	−6.8037	2.86	−10.8198	0.88
**37**	Feruloyl-*O*-sinapoylquinic acid	−12.1606	1.41	−17.0816	1.20
**50**	4″-Methyl-3-*O*-epigallocatechin gallate	−18.0349	0.97	−17.8967	1.88
**54**	Dimethoxycinnamoyl-*O*-caffeoylquinic acid	−8.1889	1.64	−12.5443	1.88
**74**	3-Caffeoylshikimic acid	−14.4907	1.57	−15.3836	1.40
**75**	Dimethoxycinnamoylquinic acid	−14.6326	1.68	−11.3988	1.69
**77**	5-*p*-Coumaroylquinic acid	−11.3266	1.65	−16.5329	1.24
**99**	Hyperoside	−9.9397	1.43	−20.6101	1.49
**100**	Isoquercetin	−10.0772	1.48	−14.7778	1.47

## Data Availability

The data is contained within the article and Appendix A.

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
