# Peer review of "Human Lung Cancer (A549) Cell Line Cytotoxicity and Anti-Leishmania major Activity of Carissa macrocarpa Leaves: A Study Supported by UPLC-ESI-MS/MS Metabolites Profiling and Molecular Docking"

_pharmaceuticals, 2022, doi:10.3390/ph15121561_

Round 1

Reviewer 1 Report

1- As authors used lung cell line just for evaluation of cytotoxicity and there is not any other evidence against Lung adenocarcinoma, I don't think so there is enough data regarding this disease to be in the title!

2- Why authors didn't use stibogluconate against L. major promastigotes to compare with test?

3- What does nil mean in the table 3? It needs explanation under the table.

4- Authors need to show the cytotoxicity against normal mammalian cells as well to show the plant derivatives are safe enough if they used lung cell line as a cancer model. 

Author Response

The responses to the reviewer comments are shown in the attached file

Reviewer 2 Report

The work has been well designed, from my point of view it is well written with very few points to correct.

For example, in line 61 you should indicate what SAR means, in table 3 what hil means? and in the bibliography lines 574 and 577 should be checked for the abbreviations of the journals

I consider the work adequate to be published in Pharmaceuticals but I should make these corrections first

Author Response

(The authors gave the same response as above.)
